# Ferrous Sulfate-Mediated Control of *Phytophthora capsici* Pathogenesis and Its Impact on Pepper Plant

**DOI:** 10.3390/plants12244168

**Published:** 2023-12-15

**Authors:** Gongfu Du, Huang He, Jiali Peng, Xiaoliang Li, Zhaohua Liu, Weixia Liu, Yan Yang, Zhiqiang Qi

**Affiliations:** Tropical Crops Genetic Resources Institute, Chinese Academy of Tropical Agricultural Sciences, Key Laboratory of Crop Gene Resources and Germplasm Enhancement in Southern China, Ministry of Agriculture and Rual Affairs, Key Laboratory of Tropical Crops Germplasm Resources Genetic Improvement and Innovation of Hainan Province, No. 4 Xueyuan Road, Longhua, Haikou 571101, China; adu209209@163.com (G.D.); hehuang_@163.com (H.H.); penglj2023@126.com (J.P.); xlli199777@163.com (X.L.); liuzhaohua123456@163.com (Z.L.); weixialiu@126.com (W.L.)

**Keywords:** *Phytophthora capsici*, ferrous sulfate (FeSO_4_), pepper, pathogen control, metabolic responses

## Abstract

*Phytophthora capsici*, a destructive fungal pathogen, poses a severe threat to pepper (*Capsicum annuum* L.) crops worldwide, causing blights that can result in substantial yield losses. Traditional control methods often come with environmental concerns or entail substantial time investments. In this research, we investigate an alternative approach involving ferrous sulfate (FeSO_4_) application to combat *P. capsici* and promote pepper growth. We found that FeSO_4_ effectively inhibits the growth of *P. capsici* in a dose-dependent manner, disrupting mycelial development and diminishing pathogenicity. Importantly, FeSO_4_ treatment enhances the biomass and resistance of pepper plants, mitigating *P. capsici*-induced damage. Microbiome analysis demonstrates that FeSO_4_ significantly influences soil microbial communities, particularly fungi, within the pepper root. Metabolomics data reveal extensive alterations in the redox metabolic processes of *P. capsici* under FeSO_4_ treatment, leading to compromised cell membrane permeability and oxidative stress in the pathogen. Our study presents FeSO_4_ as a promising and cost-effective solution for controlling *P. capsici* in pepper cultivation while simultaneously promoting plant growth. These findings contribute to a deeper understanding of the intricate interactions between iron, pathogen control, and plant health, offering a potential tool for sustainable pepper production.

## 1. Introduction

Pepper (*Capsicum annuum* L.) is a widely consumed vegetable worldwide, cherished for its culinary versatility and flavor [1]. However, the growth of pepper plants is highly susceptible to the invasive presence of *Phytophthora capsici*, a fungal pathogen notorious for causing rots and blights on various plant organs. The life cycle of *P. capsici* relies significantly on the host and encompasses both sexual and asexual reproduction. In asexual reproduction, sporangia can directly germinate, invading plant cells or producing zoospores. These zoospores are released into water, attach to plant roots, and penetrate plant cells by generating germ tubes [2,3]. For sexual reproduction, the A1 and A2 mating types undergo meiosis, resulting in the formation of oospores. These oospores then germinate to produce sporangia. These sporangia have the capacity to directly germinate, infiltrating plant cells either through enzymatic degradation or open channels [3]. Following invasion, they can spread to aerial plant parts, initiating new hyphae, sporangiophores, sporangia, and initiating a fresh cycle of infection [4]. The challenge in combatting *P. capsici* arises from its ability to produce long-lasting spores, its wide range of susceptible host plants, and its rapid dissemination through water sources [5]. Current methods employed to manage this pathogen include labor-intensive practices such as crop rotation, biofumigation, and the application of fungicides [5,6]. However, these approaches often come with the risk of environmental pollution and can be time-consuming. Alternatively, the breeding of resistant pepper cultivars is another strategy, but it entails a substantial investment of time and may result in yield losses [5]. Given the limitations of these existing methods, there is a pressing need to discover a more convenient and effective means of controlling *P. capsici*, one that can alleviate the challenges posed by this persistent and destructive fungal species.

Iron is the fourth most common element in the lithosphere and plays a critical role in plant growth and metabolism [7,8]. Fe (II) and Fe (III) are common forms of iron in nature but only Fe (II) can be absorbed by plants [8]. In plant cells, Fe is necessary for biosynthesis of chlorophyll, electron transport complexes and photosynthetic apparatus. A lack of iron leads to interveinal chlorosis in young leaves, stunted root growth, reduced yield and resistance to bio-tic and abiotic stresses [9,10], whereas excess iron may also be toxic and induce oxidative stress [11]. Moderate application of ferrous sulfate (FeSO_4_) to plants is able to promote seed germination, plant growth and antioxidant capacity [12,13]. Previous studies shown that Fe is associated with plant immunity. For examples, low-iron gives rise to hypersensitive reaction to *Xanthomonas campestris* pv. *campestris* (*Xcc*) in cabbage [14]. Similarly, the infection of *Colletotrichum graminicola* was inhibited in maize when iron was sufficient [15]. However, there was exceptions shown that in *Arabidopsis* the resistance to *Dickeya dadantii* and *Botrytis cinerea* was enhanced under iron-deficient conditions [16]. To date, little is known about the effect of iron on the immunity of pepper.

In this study, we found that FeSO_4_ inhibited the mycelial growth of *P. capsici*, enhanced the defense against *P. capsici* and improved the biomass in pepper. Microbiome analysis revealed that ferrous sulfate had a pronounced impact on the structural composition and diversity of soil microbiota in pepper root systems, particularly with respect to fungi. Furthermore, our metabolomics data illustrated that the addition of ferrous sulfate had a wide-ranging influence on the redox metabolic processes of *P. capsici*. Subsequent analysis indicated that the cell membrane permeability was compromised, and oxidative stress was induced in *P. capsici*. Consequently, we propose that FeSO_4_ represents an effective chemical solution for the control of *P. capsici* in pepper cultivation without any detrimental impact on crop yield.

## 2. Results

### 2.1. FeSO_4_ Inhibits the Growth and Virulence of P. capsici

To clarify the effect of ferrous sulfate on the growth of *P. capsici*, we conducted mycelial growth assay on culture media with different concentrations (0, 28, 56, 112 mg/L) of ferrous sulfate. The results demonstrated that ferrous sulfate inhibited the mycelial growth of Phytophthora capsica in a dose-dependent manner. After 8 days of cultivation, the best inhibitory effect on *P. capsici* mycelial growth was observed at a concentration of 112 mg/L, with an inhibition rate of 47.50%. The inhibitory effect at this concentration showed a significant difference compared to that of at other concentrations, which was 30.35% for 56 mg/L and 19.02% for 28 mg/L (Figure 1a). We also observed that the mycelia of *P. capsici* exhibited a serrated appearance under 112 mg/L ferrous sulfate treatment in comparison to that of the mock treatment, suggesting the normal branching of mycelia was suppressed (Figure 1b,c).

Inhibitory growth and abnormal morphology of the mycelia suggest that the pathogenicity of *P. capsici* is also affected. To test this hypothesis, we inoculated uniformed pepper leaves with PDA plugs from the actively growing edge of *P. capsici* grown under darkness for 8 days at with different concentrations of ferrous sulfate (0, 28, 56,112 mg/L, named ck, F28, F56, F112, respectively). The results showed that the spot size on the leaf was smaller and the inhibitory effect was stronger with increased concentration of FeSO_4_, indicating that the pathogenicity of *P. capsici* was suppressed.

We wondered the viability of leaf cells and stained the leaf samples with trypan blue four days post inoculation. Dead cells have compromised cell membranes, resulting in increased permeability, allowing trypan blue to enter and stain the cells blue. The extent of blue staining is indicative of the severity of cell death. The results revealed varying degrees of blue-stained dead cells on the pepper leaves. The CK exhibited the largest area of blue staining, measuring 3.98 cm^2^, which was significantly different from that of F28, F56 and F112. In contrast, the F112 treatment group showed the smallest area of blue staining, measuring only 0.77 cm^2^, with a reduction of 80.65% in damage compared to the control (Figure 2).

### 2.2. FeSO_4_ Improves Biomass and Resistance against P. capsici in Pepper

Iron is a micronutrient and plays an important role in chlorophyll synthesis and chloroplast structure maintenance [17]. Recent studies found that application of FeSO_4_ improves the growth of maize and rice [12,13]. To determine the effect of *P. capsici* on growth effects in pepper, we irrigated the 4-leaf-stage pepper with 200 mL 112 mg/L of FeSO_4_ solution 7 days and 12 days after transplanting, with irrigating the same volume of water as control. After irrigating twice, the plants from both of groups were inoculated with P. capsici or water. Those treated with FeSO_4_ and *P. capsici* referred to as FY; those only treated with FeSO_4_ referred to as F; those only treated with *P. capsici* referred to as Y and those treated with water referred to as CK. Biomass of the root and shoot of the pepper plants were determined 9 days after inoculating. The results showed that FeSO_4_ significantly promoted the biomass of both the dry weight and fresh weight of the shoots, although *P. capsici* had a negative effect (Figure 3a,b). However, FeSO_4_ did not promote the biomass of roots under normal conditions, but significantly promoted the biomass of roots when inoculated with *P. capsici* (Figure 3c,d). These findings suggested that the application of ferrous sulfate had a positive effect on promoting the growth of pepper plants.

We asked whether irrigating with FeSO_4_ enhances the resistance to Phytophthora capsici, and calculated the incidence rate and the disease severity index. The results showed that the incidence rate of both group FY and group Y reached 100%. However, the disease severity index was 98.00% for group Y and 46.67% for group FY, suggesting that the application of ferrous sulfate can effectively mitigate the severity of *P. capsici* in pepper plants (Figure 3e,f).

### 2.3. Global Shift in Pepper Plant Root Fungal Diversity Due to FeSO_4_ Treatment

16S and ITS rRNA sequencing (Internal Transcribed Spacer ribosomal RNA sequencing) are common amplicon sequencing methods used to identify and compare the microbial populations in each sample, be it bacteria or fungi. In order to investigate changes in the soil microbial community composition after the application of ferrous sulfate, we conducted ITS and 16S rRNA sequencing on 12 soil samples, which included four treatment groups (CK, F, FY, and Y) with three biological replicates each. Through Venn diagrams of Operational Taxonomic Units (OTUs), we explored the similarity in OTU composition between the different treatment groups. In the ITS sequencing, we detected 643 (CK), 796 (F), 919 (FY), and 677 (Y) OTUs in the soil samples. Among these, there were 304 OTUs shared between the treatment groups (Figure 4a). In the 16S rRNA sequencing, we detected 4552 (CK), 3839 (F), 4854 (FY), and 4587 (Y) OTUs in the soil samples. Among these, there were 2234 OTUs shared between the treatment groups (Figure 4b). Through the Venn diagram based on ITS and 16S rRNA sequencing, we can distinctly observe significant differences in microbial distribution between the infected group (FY and Y) and the non-infected group (CK and F). Additionally, the impact of the application of FeSO_4_ on microbial distribution is evident to some extent.

Furthermore, the coverage of each sample library exceeded 98.7%, ensuring that the study results accurately represent the microbial composition within the samples. The Chao1 index reflects the richness of the community within a sample, where a higher Chao1 value suggests greater community richness. The Shannon index and the Simpson index both serve as indices for α-diversity [18,19], with a higher Shannon value indicating greater microbial diversity, and a higher Simpson index, suggesting a more even distribution of diversity. Analysis of fungal α-diversity shows that the Chao1 value and Shannon index of the soil microbial communities significantly increased in the groups treated with ferrous sulfate (F and FY) when compared to the untreated CK and Y groups, while the Simpson index decreased significantly, indicating an increase in fungal microbial diversity following the application of ferrous sulfate (Table 1). Bacterial α-diversity analysis reveals no significant differences in the Chao1 index, the Shannon index, or the Simpson index between the groups treated with and without ferrous sulfate (i.e., CK vs. F, Y vs. FY), suggesting that the impact of ferrous sulfate on soil bacterial microbial diversity is minimal (Table 1). These results suggest that the influence of ferrous sulfate on the pepper root microbiota appears to be primarily focused on fungal diversity, with little impact on bacterial diversity.

### 2.4. FeSO_4_-Induced Significant Differences in Redox-Related Metabolites

In order to gain deeper insights into the impact of ferrous sulfate (FeSO_4_) on the metabolic profile of *P.capsici*, we conducted LC–MS untargeted metabolomics analysis on mycelia samples of the pathogen from four different treatments (each with three replicates). In total, 1183 metabolites were identified, comprising 693 in the positive ion mode and 490 in the negative ion mode. These metabolites were categorized into 18 classes, including lipids and lipid-like molecules (324, 29.56%), organic acids and derivatives (298, 27.19%), organoheterocyclic compounds (126, 11.50%), organic oxygen compounds (100, 9.12%), benzenoids (81, 7.39%), phenylpropanoids and polyketides (46, 4.20%), nucleosides, nucleotides, and analogues (36, 3.28%) (Appendix A). To assess the reliability of our data, we employed Partial Least Squares Discriminant Analysis (PLS-DA), which demonstrated good consistency between biological replicates for different time points (4d, 6d, 8d). Moreover, distinct clustering was observed among the treatment groups, and they were clearly separated from the control group (CK), indicating significant metabolic differences (Appendix A). Overall, the PLS-DA results revealed variations in metabolites between treatment groups at different time points, with the most significant differences observed between the F112 and the CK group. Given these findings, we proceeded with metabolite differential analysis and clustering analysis specifically for samples from the F112 and CK groups at different time points.

Differential accumulation of metabolites was further investigated by selecting metabolites that exhibited fold changes (FC) greater than or equal to 1 or less than or equal to 0.5 and variable importance in projection (VIP) values exceeding 1. The results of this selection were visualized in volcano plots (Figure 5a, Appendix A). We identified 141, 178, and 117 differentially accumulated metabolites in the F112 group compared to the CK group at 4, 6, and 8 days, respectively (Figure 5b, Appendix A).

The differentially accumulated metabolites are involved in various pathways, as revealed through annotation and pathway enrichment analysis using the KEGG database (Figure 5c, Appendix A). These results indicated that the differentially accumulated metabolites were primarily enriched in pathways such as glycerophospholipid metabolism, linoleic acid metabolism, arachidonic acid metabolism, tryptophan metabolism, nucleotide metabolism, cell cycle, and autophagy. Notably, glycerophospholipids are abundant in organisms and serve as crucial components of cell membranes, with the choline-containing glycerophospholipid, glycerophosphocholine, being a major osmolyte, influencing cell membrane permeability [20]. The differential expression of metabolites such as 9-HpODE, a product of long-chain lipid peroxidation, which is involved in the metabolism of linoleic acid [21], showed increased oxidative stress and antimicrobial activity. Additionally, the upregulation of S-adenosylmethionine (SAM), a vital molecule in various cellular metabolic processes, can serve as a methyl donor for over 100 different methyltransferase-catalyzed reactions, and it is a precursor in the synthesis of glutathione (GSH) [22], ultimately increasing lipid peroxidation. These findings highlight the extensive impact of ferrous sulfate on the redox processes of *P. capsici*.

### 2.5. FeSO_4_ Impact on Cell Membrane Permeability and Antioxidant Capacity

Our above analysis of the metabolome revealed that the application of ferrous sulfate (FeSO_4_) extensively influences the redox processes of *P. capsici*. However, its physiological or cellular effects remained unclear. To determine the permeability of cell membranes of under the influences by FeSO_4_, the relative electrical conductivity of cell membranes in *P. capsici* was examined after 3 days of incubation following sampling from 5-day-old PDA slugs with *P. capsici*. The results (Figure 6a) showed that the relative electrical conductivity of *P. capsici* cells increased overall with the increasing concentration of ferrous sulfate. This indicated that the cell membranes of *P. capsici* were damaged under FeSO_4_, and the extent of damage intensified as the concentration of ferrous sulfate increased.

Cell membranes are structured by lipid bilayers so that the destruction of lipids will impair the integrity of the cell membrane. Malondialdehyde (MDA) is one of the primary products of lipid peroxidation in cells under stress conditions and is an important indicator for assessing the extent of lipid peroxidation [23]. In order to understand the abundance of MDA under FeSO_4_ treatment, we measured the MDA content of *P. capsici* incubated in PDB medium after the addition of different concentrations (0, 28, 56, 112 mg/L) of FeSO_4_.

As shown in Figure 6b, under mock treatment (CK), the MDA content in *P. capsici* remained relatively constant for 72 h. However, when treated with ferrous sulfate at concentrations of 28, 56, and 112 mg/mL (F28, F56, F112), there was a rapid increase in MDA content within the first 48 h. This indicates that *P. capsici* cells experienced a significant increase in oxidative stress, leading to lipid peroxidation during this period. Nevertheless, MDA content decreased during the 48 to 72 h after FeSO_4_ treatment, which is likely due to the death of *P. capsici* cells and abolished oxidative stress response.

## 3. Discussion

Plants establish a group of microbiota that are highly associated with productivity and health, in which beneficial, neutral and pathogenic microorganisms are included [24]. The community of microbiota are dynamically affected by chemokines that selectively choose the colonized microorganism. Some chemokines guide the microbiota by affecting iron mobility and uptake of plants, highlighting the role of iron in plant–microbial interaction [25,26]. When challenged with pathogens, plants will overaccumulate iron in order to elicit ROS burst to suppress the pathogens [27]. In this study, we found that the fungal diversity of pepper root was remarkably altered when supplementing FeSO_4_ to the soil, which would be partially caused by the signals from the changing iron levels in plants. Further studies on the expression patterns of iron homeostasis and immunity genes in pepper and the tolerance of their microbiota to iron may help us understand why the diversity of microbiota was changed.

Pepper phytophthora blight is the major disease affecting peppers, and it can significantly reduce pepper yields by up to 50% [6]. High temperature and humidity, as well as excessive irrigation, all contribute to the occurrence of Phytophthora blight. Phytophthora blight spores have a long lifespan and can easily spread in the soil with flowing water, making the disease challenging to control [4]. Current strategies for the management of Phytophthora blight primarily involve field practices, such as irrigation management, crop rotation, soil solarization, and fungicide application, as well as the breeding of disease-resistant varieties. However, the use of fungicides can lead to resistance and environmental pollution, while breeding resistant varieties entails a lengthy breeding cycle. FeSO_4_ is an iron fertilizer, and researches have revealed that applying a specific concentration of FeSO_4_ in certain crops can stimulate growth and enhance the crops’ antioxidant capacity [12,28,29]. However, whether FeSO_4_ can promote the growth of peppers and alleviate Phytophthora blight has not been reported. We have observed that irrigating peppers with FeSO_4_ not only increases pepper biomass but also mitigates the damage caused by phytophthora blight. Additionally, FeSO_4_ is cost-effective, making it a novel strategy for controlling Phytophthora blight in peppers.

FeSO_4_ have been reported to display anti-fungal activity in grapevine, with different fungal pathogens exhibit different susceptibility [30]. In their study, it was found that 5 mM FeSO_4_ was required to inhibit over 50% of the pathogens. In our research, an inhibition of phytophthora was achieved with just 112 mg/L (approximately 0.7 mM) of FeSO_4_, which is more efficient. In addition, ferrous and ferric ions, have also been reported to participate in the UV-assisted Fenton reaction, which catalyzes the generation of hydroxyl radicals from hydrogen peroxide, facilitating the degradation of organic compounds and the eradication of microorganisms in wastewater. In a case study, the combination of 10 mg/L of H_2_O_2_ and 5 mg/L of Fe^3+^ exhibits the highest antimicrobial activity against *P. capsici* in distilled water [31]. In that study, iron serves as a catalyst, facilitating the generation of hydroxyl radicals, which are responsible for the actual antimicrobial activity against the zoospores of *P. capsici* in water. However, there have been no reported applications to existing sporangia or oospores present in plants or soil. Nonetheless, in pepper farming, implementing our proposed method might enhance plant immunity against *P. capsici*. Combining the photo-Fenton reaction to eliminate zoospores in irrigation water could potentially further mitigate the threat posed by *P. capsici*.

Plant–microbial interaction usually correlates with the change in redox status. When infected by pathogens, plants may induce a ROS burst to resist against the pathogen. In addition, other key molecule pairs such as reduced/oxidized glutathione (GSH/GSSG) and cysteine (Cys/CySS), and the ascorbic/dehydroascorbic acid couple (ASC/DHA) also serve as redox signals [32]. Notably, iron is able to donate or accept electron, making it active in participating redox reactions [33]. By the metabolome profiling of *P. capsici*, metabolites related to the redox state were found to be differentially accumulated. However, we did not know whether and which of these metabolites are affected by ferrous sulfate directly or indirectly. 9-HpODE, one of the members of phyto-oxylipins which come from the oxidation of polyunsaturated fatty acids, was identified as an accumulated metabolite upon *P. capsici* infection, implying a role of 9-HpODE in *P. capsici*. Similarly, soybean cultivars hyposensitive to accumulated more 9-HpODE than that in susceptible cultivars [34], suggesting the potentially shared anti-fungal mechanisms in different plant species.

Overall, this study identified FeSO_4_ as an effective chemical to manage *P. capsici* in pepper, which may function through interfering plant–microbial interaction, to affect the redox state in *P. capsici*. More importantly, application of FeSO_4_ also promotes the biomass of pepper. Therefore, our study provides an alternative in *P. capsici* control, and the application in other pathogens or plants can be expected. Further investigation into the absorption, signal transduction of FeSO_4_ in both Phytophthora and pepper would unveil the genetic network of Phytophthora resistance responding to FeSO_4_ and contribute to the breeding of the resistant germplasm of *Solanaceae*.

## 4. Materials and Methods

### 4.1. Determination of Ferrous Sulfate Antimicrobial Activity

Sterilized PDA (potato dextrose agar) was taken and heated in a microwave oven until completely melted. Once the PDA had cooled to approximately 50 °C, a specified amount of ferrous sulfate was added to prepare culture media with concentrations of 0, 28, 56, and 112 mg·L^−1^ of ferrous sulfate. Fungal strains, cultured for 5 days, were transferred from the edge of the culture to the ferrous sulfate-containing culture media, and the Petri dishes were sealed and inverted in a culture chamber maintained at (26 ± 2) °C. Culture media with an equal volume of sterile water served as a control. Three replicates were prepared for each treatment. The colony diameter was measured using a cross-streak method on days 2, 4, 6, and 8 after culturing, and photographs were taken to record the size of the colonies. The mycelial growth inhibition rate (%) was calculated as [(control colony diameter—treatment colony diameter)/(control colony diameter—diameter of the inoculum)] × 100.

### 4.2. Morphological Observation of Phytophthora Capsici Mycelium

Sterilized PDA (potato dextrose agar) culture medium was prepared with a concentration of 112 mg·L^−1^ of ferrous sulfate, and an equal volume of sterile water was added for the control. Using a hole puncher (d = 7 mm), mycelial plugs cultured for 5 days were inoculated onto each of the culture media. The plates were then incubated at (26 ± 2) °C for 8 days. The mycelial growth morphology was observed under a microscope, and photographs were taken for documentation.

### 4.3. Cell Death Staining

Cell death staining was performed following the method described by Bai et al. Pepper leaves infected with *P. capsici*, cultured in media with different concentrations of ferrous sulfate (0, 28, 56, and 112 mg·L^−1^), were used. After 4 days of infection, the infected pepper leaves were stained in boiling staining solution (100 mL lactic acid, 100 mL glycerol, 100 g phenol, and 100 mg trypan blue dissolved in 100 mL distilled water) for 5 min. Subsequently, they were destained in a 2.5 g/mL aqueous solution of hypochlorite for 24 h. Finally, photographs were taken to document the leaf samples.

### 4.4. Measurement of Relative Electrical Conductivity

Colletotrichum fungi cultured for 5 days on PDA were collected using a hole puncher (d = 7 mm) and placed into PDB (Potato Dextrose Broth) culture media with varying concentrations of ferrous sulfate (28, 56, 112 mg·L^−1^), with an equal volume of sterilized ddH_2_O added as a negative control. After 3 days of incubation on a shaker at 150 r·min^−1^ at 25 °C, the electrical conductivity of the culture solution was measured using a conductivity meter. The culture solution was then heated to boiling, allowed to cool, and the electrical conductivity was measured again. Relative electrical conductivity (%) was calculated as (conductivity of the culture solution/conductivity after heating) × 100.

### 4.5. Determination of Malondialdehyde (MDA)

Following the method by Shao et al. (2013) [35] as a reference, mycelial plugs (d = 7 mm) from 5-day-old *P. capsici* colonies were inoculated into PDB (Potato Dextrose Broth) culture media. The cultures were incubated for 48 h at 160 r·min^−1^ and 25 °C. Ferrous sulfate was added to the PDB culture media at a concentration of 112 mg·mL^−1^, with sterilized ddH_2_O added as a control. After 72 h of incubation, the MDA (malondialdehyde) content was determined. For MDA determination, 0.5 mL of MDA extraction solution was mixed with 0.5 mL of 0.67% mass concentration of thiobarbituric acid. The mixture was heated for 15 min and then cooled in cold water, followed by centrifugation at 5000× *g* for 10 min. The supernatant was adjusted to zero using a 5% trichloroacetic acid solution, and the absorbance values (A) were measured at 450, 532, and 600 nm. The formula for calculating MDA concentration in micromoles per liter (μmol·L−1) is as follows: MDA concentration (μmol·L^−1^) = 6.45(A532 nm − A600 nm) − 0.56A450 nm.

### 4.6. Disease Resistance and Biomass Measurement

Pepper seedlings at the four-leaf stage with consistent size were transplanted into pots. Four treatments, each with 30 plants, were set up with three repetitions. Treatment 1: distilled water control (CK). Treatment 2: ferrous sulfate 112 mg/L (F). Treatment 3: ferrous sulfate + *P. capsici* (FY). Treatment 4: *P. capsici* 106/mL (Y). After acclimatizing the seedlings for 7 days, Treatments 2 and 3 were irrigated with 200 mL of sterile water containing 112 mg/L of ferrous sulfate per plant. Treatments 1 and 4 were irrigated with an equal amount of sterile water, and this root irrigation was performed every 5 days for two consecutive times. After the second root irrigation with ferrous sulfate, Treatments 3 and 4 were inoculated with *P. capsici*, with each plant receiving 10 mL of a suspension containing 106/mL of motile spores. Treatments 1 and 2 received an equal amount of sterile water. Twenty-one days after transplantation, disease incidence and biomass were measured.

### 4.7. Metabolome and Soil Microbial Community Analysis

For mycelial metabolome analysis, inoculate mycelial plugs (7 mm in diameter) from 5-day-old fungal colonies at the edge of the culture onto PDB (Potato Dextrose Broth) culture media containing a certain amount of ferrous sulfate to prepare culture media with concentrations of 0, 28, 56, and 112 mg·L^−1^ of ferrous sulfate. Seal the culture and incubate them on a shaker at (26 ± 2) °C and a speed of 105 r/min. Use PDB culture media with an equal amount of sterilized ddH_2_O as a control. Harvest mycelial balls at 4 days, 6 days, and 8 days and store them at −80 °C. Repeat this process biologically three times.

For soil microbial community analysis of pepper roots, nine days after inoculation, collect soil from the pepper roots for four treatments (CK, F, FY, Y), with three repetitions for each treatment. Combine the soil from two pepper root samples for each repetition. The samples are sieved to 2 mm, bagged, and stored at −80 °C.

### 4.8. Data Processing

Variance analysis was conducted using SPSS software (v25). Pepper root-associated soil microbial diversity analysis was performed using the cloud data analysis platform I-sanger provided by Shanghai Meiji Biomedical Technology Co., Ltd. (Shanghai, China). A species composition community graph was constructed. Microbial community diversity indices, including the Chao 1 index, the Shannon index, and the Simpson index, were calculated. The Chao 1 index is used to measure species richness, with larger values indicating higher species richness. The Shannon index is used to measure species diversity, and the Simpson index represents the probability that randomly sampled individuals in the community belong to the same species, also used to measure species diversity. A higher Shannon index and a lower Simpson index indicate higher species diversity in the sample.

## 5. Conclusions

In summary, this study highlights the potential of ferrous sulfate as a promising method to control *P. capsici*, a destructive plant pathogen that affects pepper cultivation. FeSO_4_ not only inhibits the growth and virulence of the pathogen but also enhances the biomass of pepper plants. This approach could provide a more convenient and environmentally friendly alternative to traditional disease management strategies.

## Figures and Tables

**Figure 1 plants-12-04168-f001:**
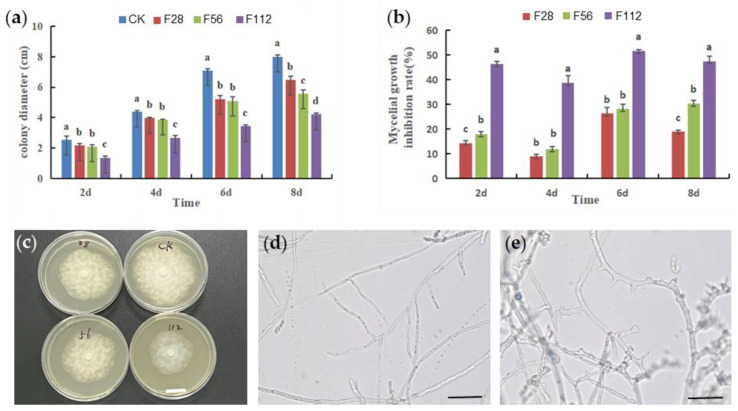
The inhibitory effect of FeSO_4_ on *P. capsici* growth at different concentrations. (**a**) The colony diameter of *P. capsici* under the treatment of FeSO_4_ for 2, 4, 6 and 8 days. CK, mock treatment; F28, 28 mg/L; F56, 56 mg/L, F112, 112 mg/L. Data are presented as means ± SD. Letters on histograms indicate significant differences according to ANOVA following the least-significant difference (LSD) test (*p* < 0.05). (**b**) The inhibition rate of mycelial growth of *P. capsici* under the treatment of FeSO_4_ for 2, 4, 6 and 8 days. Data are presented as means ± SD. Letters on histograms indicate significant differences according to ANOVA following the least-significant difference (LSD) test (*p* < 0.05). (**c**) A view of *P. capsici* grown on plates with different concentrations of FeSO_4_. (**d**,**e**) the morphology of mycelial of *P. capsici* with (**e**) and without (**d**) 112 mg/L FeSO_4_ (Scale bar = 50 μm).

**Figure 2 plants-12-04168-f002:**
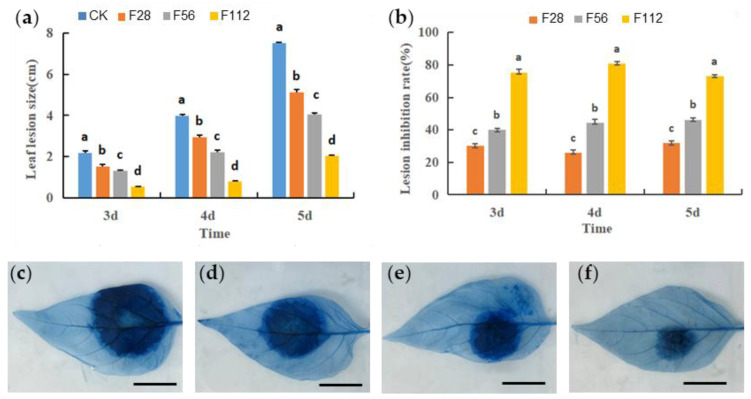
The effect of FeSO_4_ on the virulence of *P. capsici* at different concentrations. (**a**,**b**) Leaf lesion size (**a**) and lesion inhibition rate (**b**) under the treatment of FeSO_4_ for 3, 4 and 5 days. CK, mock treatment; F28, 28 mg/L; F56, 56 mg/L, F112, 112 mg/L. Data are presented as means ± SD. Letters on histograms indicate significant differences according to ANOVA following the least-significant difference (LSD) test (*p* < 0.05). (**c**–**f**) Trypan blue staining of pepper leaves inoculated with *P. capsici* grown without (**c**) or with 28 mg/L, (**d**) 56 mg/L (**e**) and 112 mg/L (**f**) of FeSO_4_ (Scale bar = 1 cm.).

**Figure 3 plants-12-04168-f003:**
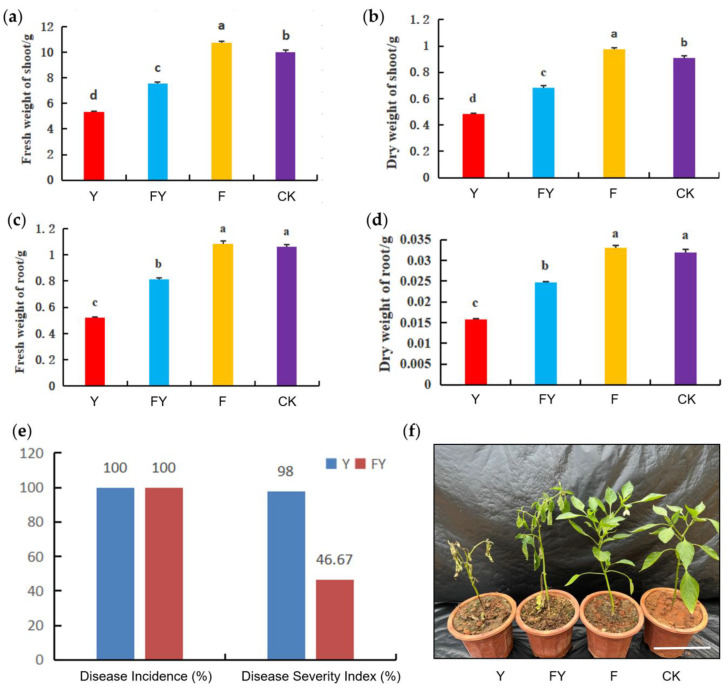
Biomass and resistance pepper plants in response to FeSO_4_ and *P. capsici*. (**a**,**b**) Fresh (**a**) weight and dry weight (**b**) of the shoots in response to FeSO_4_ and Phytophthora capsici. Data are presented as means ± SD. Letters on histograms indicate significant differences according to ANOVA following the least-significant difference (LSD) test (*p* < 0.05). (**c**,**d**) Fresh weight (**c**) and dry weight (**d**) of the roots in response to FeSO_4_ and *P. capsici*. Data are presented as means ± SD. Letters on histograms indicate significant differences according to ANOVA following the least-significant difference (LSD) test (*p* < 0.05). (**e**) The disease incidence rate and the disease severity index caused by *P. capsici*. (**f**) Growth status of pepper plant under different treatments. (Scale bar = 15 cm.) CK, mock treatment; F, plants were irrigated with 200 mL of 112 mg/L FeSO_4_ twice; FY, plants were irrigated with 200 mL of 112 mg/L FeSO_4_ twice, following by inoculated with *P. capsici*; Y, plants were inoculated with *P. capsici*.

**Figure 4 plants-12-04168-f004:**
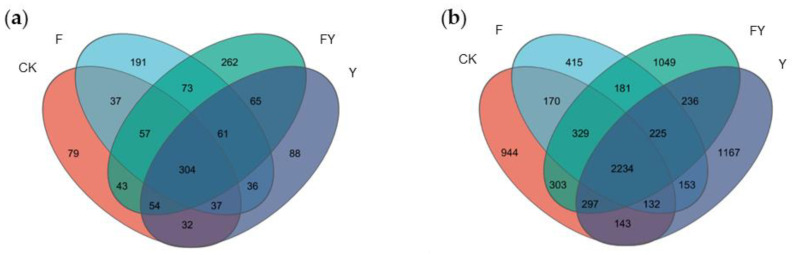
Diversity analysis of pepper root microbial community. (**a**) Gene outs distribution of fungal based on OTUs. (**b**) Gene outs distribution of bacteria based on OTUs.

**Figure 5 plants-12-04168-f005:**
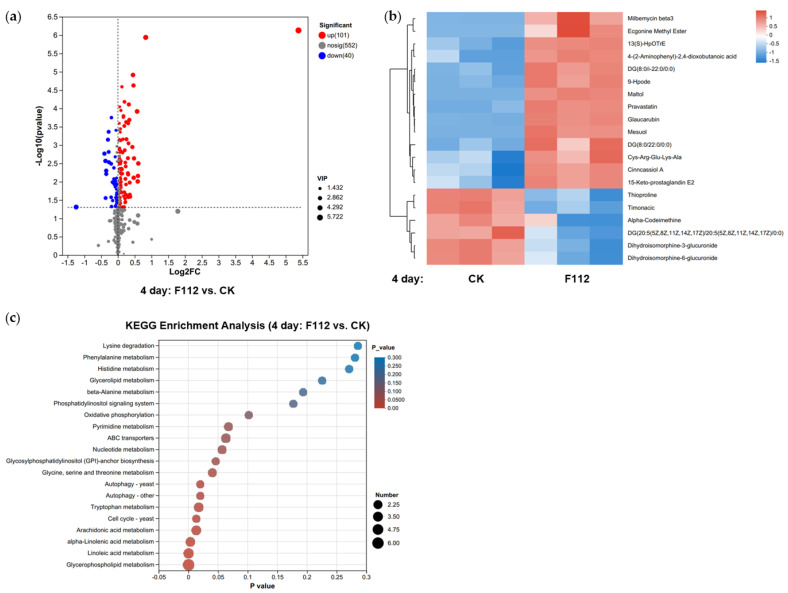
Clustering analysis of differentially expressed metabolites between the F112 and CK groups at 4 days. (**a**) Volcano plots depicting differentially expressed metabolites between the F112 and CK groups at 4 days. The horizontal axis represents the log2 fold change (log2 FC) in metabolite expression between the two groups, while the vertical axis represents the statistical significance of differences in metabolite expression (−log10(*p*_value)). Larger points indicate higher variable importance in projection (VIP) values. Points on the left side represent downregulated metabolites, while points on the right side represent upregulated metabolites. The further to the left or right, and higher up on the plot, the more significant the expression difference. (**b**) Hierarchical clustering dendrograms illustrating the clustering of differentially expressed metabolites between the F112 and CK groups at 4 days. Closer branches indicate similar expression patterns for all metabolites within the samples. Each column represents a sample, with sample names shown below. Each row represents a metabolite, with color indicating the relative expression level in the respective group, and the color gradient is depicted in the gradient color bar. (**c**) Pathway enrichment plots for the F112 and CK groups at 4 days. The horizontal axis represents the significance *p*-value of the enrichment, with lower *p*-values indicating greater statistical significance (*p* < 0.05 indicates significant enrichment). The vertical axis represents KEGG pathways. The size of bubbles in the plot represents the degree of enrichment of compounds in that pathway.

**Figure 6 plants-12-04168-f006:**
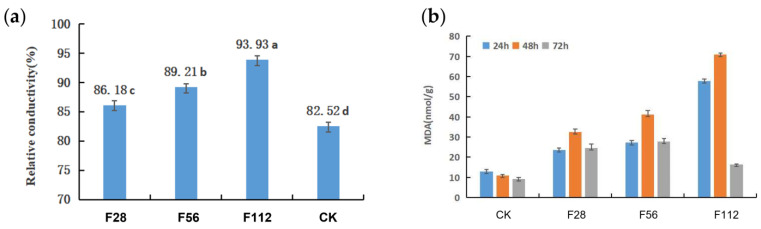
Effects of FeSO_4_ on *P. capsici* cell membrane permeability and lipid peroxidation. (**a**) Effects of FeSO_4_ with different concentrations (CK, 0; F28, 28 mg/L; F56, 56 mg/L; F112, 112 mg/L) on the relative conductivity of *P. capsici*. Data are presented as means ± SD. Letters on histograms indicate significant differences according to ANOVA following the least-significant difference (LSD) test (*p* < 0.05). (**b**) Changes in MDA content at different times after FeSO_4_ with different concentrations (CK, 0; F28, 28 mg/L; F56, 56 mg/L; F112, 112 mg/L).

**Table 1 plants-12-04168-t001:** Effect of different treatments on the soil microbial diversity index.

**Microbe**	**Group**	**Chao1 Index**	**Shannon Index**	**Simpson Index**	**Coverage**
Fungi	CK	433.55 ± 14.96 b	1.78 ± 0.07 b	0.48 ± 0.03 a	0.998
	F	497.62 ± 44.05 ab	3.30 ± 0.12 a	0.14 ± 0.02 b	0.999
	FY	566.37 ± 64.06 a	3.70 ± 0.31 a	0.07 ± 0.02 c	0.999
	Y	431.56 ± 65.71 b	1.67 ± 0.08 b	0.52 ± 0.03 a	0.999
Bacteria	CK	3165.69 ± 140.28 ab	5.92 ± 0.13 bc	0.011 ± 0.0020 a	0.987
	F	2987.87 ± 18.23 b	5.88 ± 0.02 c	0.010 ± 0.0009 a	0.987
	FY	3367.95 ± 239.63 a	6.13 ± 0.16 a	0.008 ± 0.001 ab	0.987
	Y	3106.20 ± 180.22 ab	6.10 ± 0.06 ab	0.007 ± 0.0004 b	0.988

Data are presented as means ± SD. Letters on histograms indicate significant differences according to ANOVA following the least-significant difference (LSD) test (*p* < 0.05).

## Data Availability

All data supporting the findings of this study are available within the article and its Appendix A.

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
