# Peer review of "Ferrous Sulfate-Mediated Control of Phytophthora capsici Pathogenesis and Its Impact on Pepper Plant"

_plants, 2023, doi:10.3390/plants12244168_

Round 1

Reviewer 1 Report

Comments and Suggestions for Authors

The manuscript gives new information about effect of FeSO4 against Phytophthora capsici on pepper plants. But I think the manuscript has some problems and it requires “major revisions” for publication.

1  the test on culture media

Authors performed growth test, virulence test and membrane permeability test on culture media, and they discuss the FeSO4 control effect against P. capsici using the results. But I think there are many differences between culture media test and in plata test.

For example, Is the morphology of mycelia different on the pepper plants between “Y” and “FY”?

2 inhibitory of growth on the leaf (Fig.2)

    Authors use P. capsici cultured on the medium for inoculation into pepper leafs. But each P. capsici is affected by high concentration of FeSO4 during culture on the medium.

    They should inoculate same P. capsici sample into the leaf and then different concentration of FeSO4 is sprayed.

3 Figure 4

    I can’t understand what authors explain using this test.

4 micro diversity index

    Why low Simpson index mean the high microbial diversity?

5 L294

    Authors discuss “which is probably caused by the signals from the changing iron levels in plants”. Is the conclusion appropriate, though they described direct effect on P. capsici using various tests?

6 others

    Authors should revise figure number during the manuscript.

Author Response

Reviewer 1

The manuscript gives new information about effect of FeSO4 against Phytophthora capsici on pepper plants. But I think the manuscript has some problems and it requires “major revisions” for publication.

1  the test on culture media

Authors performed growth test, virulence test and membrane permeability test on culture media, and they discuss the FeSO4 control effect against P. capsici using the results. But I think there are many differences between culture media test and in plata test.

For example, Is the morphology of mycelia different on the pepper plants between “Y” and “FY”?

Response: Thank you very much for your insightful comments and suggestions. Your concerns regarding Figure 1 and the subsequent question about Figure 2 essentially focus on the distinctions between culture media tests and in planta tests. In Figures 1 and 2, we employed culture media with varying FeSO4 concentrations to gain insights into the growth patterns and mycelial status of P. capsici. This approach aimed to provide preliminary understanding of FeSO4's inhibitory effects on P. capsici and identify the most potent concentration for subsequent plant-microbe interaction assessments. Through these experiments, we confirmed the inhibitory impact of FeSO4 on P. capsici, opting for the concentration (112 mg/L) with relatively superior inhibitory effects. This concentration was then applied in the subsequent Disease Resistance and Biomass Measurement experiment (Figure 3), where the same P. capsici sample was inoculated into the leaf.

We fully recognize the significance of in planta experiments for a more holistic comprehension of the interaction. Currently, we are in the process of planning and preparing these experiments, including the morphological difference of mycelia on the pepper plants and inoculation same P. capsici sample into the leaf following different concentration of FeSO4. However, due to time constraints imposed by the editorial office (a 10-day revision period), conducting and incorporating new in planta experiments within this limited timeframe is not feasible.

We appreciate your understanding of our present constraints and assure you of our commitment to conducting additional in planta experiments in future studies to enhance the overall depth and robustness of our research. Thank you once again for your constructive feedback.

2 inhibitory of growth on the leaf (Fig.2)

    Authors use P. capsici cultured on the medium for inoculation into pepper leafs. But each P. capsici is affected by high concentration of FeSO4 during culture on the medium.

    They should inoculate same P. capsici sample into the leaf and then different concentration of FeSO4 is sprayed.

Response: Thank you for your meticulous review of our manuscript. As mentioned in the preceding response, the primary objective of Figures 1 and 2 was to comprehend the growth conditions and mycelial status of P. capsici under various FeSO4 concentrations on culture media. In Figure 3, we deliberately conducted the inoculation of the same P. capsici sample into the leaf, followed by treatment with 112 mg/L FeSO4. This experimental design, elucidated in the "4.6 Disease Resistance and Biomass Measurement" section of the Materials and Methods, aimed to assess the efficacy of ferrous sulfate in mitigating P. capsici severity under conditions that more accurately emulate the in planta environment.

We appreciate your suggestion to explore the effect of different FeSO4 concentrations when sprayed after inoculation, considering it a valuable point for further investigation. In our forthcoming studies, we commit to incorporating experiments specifically addressing the impact of varying FeSO4 concentrations post-inoculation.

3 Figure 4

    I can’t understand what authors explain using this test.

Response: The Venn diagram in Figure 4 serves to illustrate the shared and unique OTU numbers among the four samples (CK, F, FY, and Y). It provides a visually intuitive representation of the similarity and overlap in OTU composition among different samples. Through the Venn diagram based on ITS and 16S rRNA sequencing in Figure 4, we can distinctly observe significant differences in microbial distribution between the infected group (FY and Y) and the non-infected group (CK and F). Additionally, the impact of the application of FeSO4 on microbial distribution is evident to some extent. We have incorporated this clarification into the revised manuscript, specifically in Lines 168-171 (marked in red). Thank you for your valuable feedback.

4 micro diversity index

    Why low Simpson index mean the high microbial diversity?

Response: We appreciate your question regarding the interpretation of microbial diversity indices, specifically the Simpson index. The Simpson index is a measure of diversity that considers both the number of species (richness) and the distribution of individuals among those species (evenness). It ranges from 0 to 1, where a lower value indicates higher diversity. In the context of the Simpson index:

  • Low values (closer to 0): This suggests high diversity. When the Simpson index is low, it means that no single species dominates the community, and the distribution of individuals among different species is more even.
  • High values (closer to 1): This suggests low diversity. A high Simpson index indicates that one or a few species dominate the community, leading to a less even distribution of individuals.

5 L294

    Authors discuss “which is probably caused by the signals from the changing iron levels in plants”. Is the conclusion appropriate, though they described direct effect on P. capsici using various tests?

Response: We acknowledge your astute observation regarding the conclusion drawn in relation to the changing iron levels in plants. The statement you highlighted indeed stems from the notion that certain chemokines, influenced by alterations in iron mobility and plant uptake, play a role in guiding the microbiota. While our various tests demonstrated a direct effect of FeSO4 on P. capsici, we recognize the complexity of interactions in the soil environment. To address this, we have revised the sentence in question to convey a more nuanced perspective. The revised statement now reads, "which would be partially caused by the signals from the changing iron levels in plants," offering a more tempered interpretation of our findings. Please see Line 300 marked in red in the revised manuscript. Thank you.

6 others

    Authors should revise figure number during the manuscript.

Response: Thank you for bringing to our attention the discrepancy in the figure numbering within the manuscript. The original manuscript incorrectly referenced Figure 3 in the Results section. In our revised version, this error has been rectified to ensure accurate and consistent figure numbering throughout the manuscript.

Reviewer 2 Report

Comments and Suggestions for Authors

The manuscript titled "Ferrous Sulfate-Mediated Control of Phytophthora Capsici Pathogenesis and Its Impact on Pepper Plant" by Gong fu Du et al. presents an interesting approach involving FeSO4 to protect pepper from its devastating pathogen, Phytophthora capsica. Microbiome analysis shows that FeSO4 significantly influences soil microbial communities, particularly fungi, inside the pepper root. Interestingly, metabolomics data presented by the authors reveal extensive alterations in the redox metabolic processes of P. capsici under FeSO4 treatment, which ultimately leads to compromised cell membrane permeability and oxidative stress in the fungus, contributing to its demise. Overall, the topic is certainly of interest to the readers of Plants and merits consideration for publication.

 Title:

- please replace Phytophthora Capsici with Phytophthora capsici

Abstract:

- informative without being too long, no criticisms

Introduction:

- please use the correct nomenclature for chemical formulae, FeSO4, not FeSO4. This should also be corrected for all other sections of the manuscript.

- please use italicized letters when giving scientific names of species (e.g., Colletotrichum graminicola, not Colletotrichum graminicola. This should also be corrected for all other sections of the manuscript.

- “Iron is the fourth largest element in the lithosphere” à large is not correct here, the authors probably mean the fourth most common element?

- please describe the biology and pathogenicity / life-cycle of Phytophthora capsica in more detail. This information is relevant for understanding some of the analyses the authors have done regarding lipid peroxidation, oxidative stress response and infection potential.

Results:

- Legend to figure 1: morphology, not morphology

- “The ck exhibited…” à Please clarify what ck means. Is it the same as CK?

- “Figure 4. Diversity analysis of pepper root microbial…” à The authors mean microbial community?

- Figure 5: The text is not readable; it is important to correct this problem because the reader might not bother to study this figure.

- Did the authors consider a Gene Ontology analysis of their metabolomics data? It could improve the results section considerably.

Discussion:

- It is important that the authors compare their results to an article that deals with a similar topic of using FeSO4 to control Phytophthora capsici (Polo-Lopéz et al., (2013) Benefits of photo-Fenton at low concentrations for solar disinfection of distilled water. A case study: Phytophthora capsici. Catalysis Today Volume 209, pages 181-187 https://doi.org/10.1016/j.cattod.2012.10.006). How do the results of the authors differ? Could there be a potential for a synergistic approach?

Materials and Methods:

- all information for the reproduction of the experiments is given

Recommendation:

- This manuscript requires major revision before it is acceptable in the journal Plants.

Comments on the Quality of English Language

Minor to moderate editing of English is recommended.

Author Response

Reviewer 2

The manuscript titled "Ferrous Sulfate-Mediated Control of Phytophthora Capsici Pathogenesis and Its Impact on Pepper Plant" by Gong fu Du et al. presents an interesting approach involving FeSO4 to protect pepper from its devastating pathogen, Phytophthora capsica. Microbiome analysis shows that FeSO4 significantly influences soil microbial communities, particularly fungi, inside the pepper root. Interestingly, metabolomics data presented by the authors reveal extensive alterations in the redox metabolic processes of P. capsici under FeSO4 treatment, which ultimately leads to compromised cell membrane permeability and oxidative stress in the fungus, contributing to its demise. Overall, the topic is certainly of interest to the readers of Plants and merits consideration for publication.

 Title:

- please replace Phytophthora Capsici with Phytophthora capsici

Response: We have revised the word “capsici” as you suggested, thank you!

Abstract:

- informative without being too long, no criticisms

Response: Thanks for the comment.

Introduction:

- please use the correct nomenclature for chemical formulae, FeSO4, not FeSO4. This should also be corrected for all other sections of the manuscript.

Response: The nomenclature for FeSO4 has been corrected throughout the text, thanks.

- please use italicized letters when giving scientific names of species (e.g., Colletotrichum graminicola, not Colletotrichum graminicola. This should also be corrected for all other sections of the manuscript.

Response: Thank you for your reminding. We have corrected the scientific names of the species by italicized letters in the text.

- “Iron is the fourth largest element in the lithosphere” à large is not correct here, the authors probably mean the fourth most common element?

Response: We are sorry for the incorrect statement here and we have rewritten the sentence as “Iron is the fourth largest most common element in the lithosphere and…”. Please see Line 53, marked in red. Thanks.

- please describe the biology and pathogenicity / life-cycle of Phytophthora capsica in more detail. This information is relevant for understanding some of the analyses the authors have done regarding lipid peroxidation, oxidative stress response and infection potential.

Response: We have incorporated additional information regarding the life cycle of P. capsici. Please see lines 34-42 that marked in red. Your input has been invaluable, thanks.

Results:

- Legend to figure 1: morphology, not morphology

Response: The word morphology has been corrected.

- “The ck exhibited…” à Please clarify what ck means. Is it the same as CK?

Response: Yes. We have replaced “ck” with capital letters.

- “Figure 4. Diversity analysis of pepper root microbial…” à The authors mean microbial community?

Response: Thank you for the comment. We have revised this in Figure 4 legend.

- Figure 5: The text is not readable; it is important to correct this problem because the reader might not bother to study this figure.

Response: Thank you for bringing this to our attention. In the revised version, we have divided the original Figure 5 into three separate figures: the new Figure 5, Supplementary Figure 2, and Supplementary Figure 3. The text size has been increased in these split images to enhance readability. We appreciate your feedback and hope this adjustment improves the clarity of the figures for readers.

- Did the authors consider a Gene Ontology analysis of their metabolomics data? It could improve the results section considerably.

Response: We appreciate your interest in enhancing the results section. Gene Ontology (GO) analysis is usually conducted for gene-level functional annotation, providing insights into the potential involvement of each gene in specific pathway terms or GO terms. In contrast, metabolomics data spans a diverse range of metabolic pathways, making Kyoto Encyclopedia of Genes and Genomes (KEGG) pathway enrichment analysis more suitable, as it addresses pathways rather than specific genes. Thank you for your valuable suggestion.

Discussion:

- It is important that the authors compare their results to an article that deals with a similar topic of using FeSO4 to control Phytophthora capsici (Polo-Lopéz et al., (2013) Benefits of photo-Fenton at low concentrations for solar disinfection of distilled water. A case study: Phytophthora capsici. Catalysis Today Volume 209, pages 181-187 https://doi.org/10.1016/j.cattod.2012.10.006). How do the results of the authors differ? Could there be a potential for a synergistic approach?

Response: Thank you for your suggestion. The added discussion can highlight the unique aspects and significance of this study. We have incorporated the discussion about the study of Polo-Lopéz et al., 2013. The newly added discussion can be seen from lines 324 to 336 that marked in red.

Materials and Methods:

- all information for the reproduction of the experiments is given

Response: Thank you for your comments.

Round 2

Reviewer 1 Report

Comments and Suggestions for Authors

The manuscript gives new information about effect of FeSO4 against Phytophthora capsici on pepper plants. I think the manuscript is acceptable in present form.

Reviewer 2 Report

Comments and Suggestions for Authors

The authors have addressed my prior comments very well. I herewith recommend acceptance of the manuscript for publication in the journal Plants.
